# Prevalence of Virulence Genes and Antimicrobial Resistances in *E. coli* Associated with Neonatal Diarrhea, Postweaning Diarrhea, and Edema Disease in Pigs from Austria

**DOI:** 10.3390/antibiotics9040208

**Published:** 2020-04-24

**Authors:** René Renzhammer, Igor Loncaric, Franz-Ferdinand Roch, Beate Pinior, Annemarie Käsbohrer, Joachim Spergser, Andrea Ladinig, Christine Unterweger

**Affiliations:** 1University Clinic for Swine, University of Veterinary Medicine, 1210 Vienna, Austriachristine.unterweger@vetmeduni.ac.at (C.U.); 2Institute of Microbiology, University of Veterinary Medicine, 1210 Vienna, Austria; loncarici@staff.vetmeduni.ac.at (I.L.);; 3Unit of Veterinary Public Health and Epidemiology, Institute of Food Safety, Food Technology and Veterinary Public Health, University of Veterinary Medicine, 1210 Vienna, Austria

**Keywords:** antimicrobial resistance, *Escherichia coli*, *fimH*, neonatal diarrhea, pigs, post-weaning diarrhea, virulence factors

## Abstract

Increasing numbers of multi-resistant *Escherichia (E.) coli* from clinical specimens emphasize the importance of monitoring of their resistance profiles for proper treatment. Furthermore, knowledge on the presence of virulence associated genes in *E. coli* isolates from European swine stocks is scarce. Consequently, a total of 694 *E. coli* isolated between 2016 and 2018 from diarrheic piglets of Austrian swine herds were investigated. The isolates were tested for their susceptibility to twelve antibiotics using agar disk diffusion test and for the presence of 22 virulence associated genes via PCR. Overall, 71.9, 67.7, and 49.5% of all isolates were resistant to ampicillin, tetracycline, and trimethoprim-sulfamethoxazole, while resistance levels to gentamicin and fosfomycin were 7.7 and 2.0%, respectively. Resistance frequency to ciprofloxacin was higher than in previous studies. Isolates were more likely to be resistant to ampicillin if they were also resistant to ciprofloxacin. No isolate was resistant to meropenem or amikacin. Virulence genes were detected more frequently in isolates expressing hemolytic activity on blood agar plates. The detection rate of *faeG* was increased in *fimH* negative isolates. We assume, that hemolytic activity and absence of *fimH* could be considered as potential indicators for the virulence of *E. coli* in piglets.

## 1. Introduction

The emergence of multi-resistant bacteria is commonly considered to be a consequence of the misuse and overuse of antibiotics [1,2]. In particular, the emergence of extended-spectrum β-lactamases (ESBL) producing *Escherichia (E.) coli* [3] has led to major concerns over the past few years. A study demonstrated that distinct numbers of pathogenic porcine *E. coli* were resistant to antibiotics that are also commonly applied in swine stocks, such as tetracyclines and trimethoprim/sulfonamide [4]. An increasing number of *E. coli* isolates resistant to cephalosporins and fluoroquinolones has also been reported repeatedly [5,6]. *Enterobacteriales* (including *E. coli, Proteus* spp., *Klebsiella pneumoniae*) being resistant to 3rd generation cephalosporins and/or carbapenems have been classified as “Priority 1: Critical group” bacterial pathogens by the World Health Organization [7].

Pathogenic *E. coli* are one of the main causative agents of diarrhea in swine. They are particularly known to cause neonatal diarrhea, as well as post-weaning diarrhea (PWD) and edema disease (ED), leading to severe economic consequences in pig stocks due to increased mortality rates and decreased growth rates [8].

In particular, Enterotoxigenic *E. coli* (ETEC) represent the largest group of *E. coli* causing diarrhea in suckling piglets and newly weaned piglets [9]. The adherence to intestinal epithelial cells and the production of certain toxins are two major conditions that facilitate ETEC to induce diarrhea. The heat-labile enterotoxin LT (*elt*) and the heat-stable enterotoxins STa and STb (*sta*, *stb*) are the best-known toxins, which possess the ability to induce diarrhea due to various changes of the electrolyte equilibrium [10]. Epithelial adherence is predominantly facilitated by adhesive fimbriae like F4 (*faeG*) and F18 (*fedA*), as well as F5 (*fanC*), F6 (*fasA*), and F41 (*fim41A*). The detection of at least one enterotoxin gene (*elt, sta, stb*), together with one gene coding for fimbriae, including F4, F5, F6, F18, and F41, in a single *E. coli* isolate is defined as an essential criterion for the classification of porcine ETEC [8]. Recently, an emerging subgroup of *E. coli* was described with *E. coli* belonging to this group not harboring any mentioned fimbrial genes, but being positive for *aidA*, a gene encoding the ‘adhesion involved in diffuse adherence-I’ (AIDA-I) autotransporter adhesin [11]. It was clearly demonstrated that *E. coli* possessing AIDA-I but no classical fimbriae (F4, F5, F6, F18, F41) were able to mediate severe colonization and biofilm formation in the intestines, thus inducing diarrhea [12]. Another virulence gene that has been associated with neonatal diarrhea and PWD is *astA*, encoding the enteroaggregative heat-stable enterotoxin 1 (EAST1) [13]. Hemolytic activity in ETEC from weaned piglets is observed in most isolates containing *faeG* and in all isolates containing *fedA*. Thus, hemolytic activity could act as an indicator for pathogenicity, although its impact should not be overestimated [14].

Besides ETEC, Enteropathogenic *E. coli* (EPEC) also play a significant role in the pathogenesis of PWD. The ‘locus for enterocyte effacement’ (LEE) represents a pathogenicity island of EPEC carrying genes that facilitate attaching and effacing lesions. One gene of LEE, *escV*, code for a type III secretion system and is commonly used as diagnostic criterion to identify EPEC [15].

Edema disease *E. coli* (EDEC) are responsible for vascular damage resulting in edema disease (ED) [14] and are determined by the presence of F18 fimbriae (*fedA*) and the Shiga toxin Stx2e.

The impact of *fimH*, one of the genes encoding Type-1 fimbriae, in the pathogenesis of neonatal diarrhea in piglets has already been doubted [16]. It is frequently described as a virulence gene of Uropathogenic *E. coli* (UPEC) together with *cnf1,* which encodes for cytotoxic necrotizing factor 1 (CNF1) and *papC*, a gene with an important role in P fimbriae assembly [17]. However, CNF1 and P fimbriae were occasionally discussed to play a role in the pathogenesis of PWD too [18].

Knowledge on the presence of virulence genes in *E. coli* isolated from Central European swine stocks is scarce. Consequently, the present study mainly aimed to investigate the frequency of certain virulence genes in *E. coli* isolates from piglets displaying diarrhea or clinical signs of ED in Austria. In addition, the resistance profiles of clinical *E. coli* isolates were determined in order to specify the current situation of antimicrobial resistances in Austrian swine stocks.

## 2. Results

### 2.1. Antimicrobial Resistance Profiles

Resistances to ampicillin were predominant (71.9%), followed by resistances to tetracycline (67.7%) and trimethoprim-sulfamethoxazole (49.5%), whereas none of the isolates was resistant to meropenem or amikacin (Table 1). Approximately 2% of all isolates were resistant to fosfomycin and aztreonam. Among tested antibiotics that are regularly applied to treat colibacillosis in swine stocks, resistances against gentamicin (7.7%) were the least common (Table 1). In total, 16.4% of all isolates were resistant to ciprofloxacin. Isolates that were resistant to ciprofloxacin were more likely to be resistant to ampicillin (OR = 5.381 (95% CI 2.855–10.140)) and trimethoprim-sulfamethoxazole (OR = 3.482 (95% CI 2.554–4.747)). Compared to non-hemolytic *E. coli*, isolates with hemolytic activity showed higher resistance rates to trimethoprim-sulfamethoxazole, chloramphenicol, and gentamicin. Isolates that were determined as ESBL-producing *E. coli* were more frequently resistant to eight out of those ten tested antibiotics, for which resistances were observed, than non-ESBL-producing *E. coli* (Table 1). Out of all ceftazidime resistant isolates, 88.2% could be classified as ESBL-producing *E. coli*. Generally, 27.1% of all isolates from suckling piglets and 16.2% of all isolates from weaned piglets were classified as ESBL-producing *E. coli*. No ESBL-producing isolate showed hemolytic activity.

Weaned piglets had significantly lower resistance rates to tested antibiotics in all three calculated generalized linear models (Model 1: *p* = 0.0107, Model 2: *p* = 0.0397, Model 3: *p* = 0.0019). Examined isolates, originating from swine stocks supervised by herd veterinarian 1 exhibiting the lowest resistance rates to tetracycline, had the second highest resistance rates to ciprofloxacin (Figure 1). However, considering age group and clinical symptoms, the data show no significant influence of the herd veterinarian on the resistance profiles in any of the generalized linear models.

### 2.2. Virulence Genes

Out of 694 tested isolates, 39 did not carry any of 22 examined virulence genes, while *fimH* was detected most frequently compared to other virulence genes (552/694). Due to the uncertainty about the impact of several virulence factors in the pathogenesis of neonatal diarrhea, PWD and ED only those which are known to be associated with ETEC or EDEC are visualized (Table 2 and Table 3) and further discussed in more detail. Only a small number of isolates contained genes coding for F5 (8/694), F6 (3/694), and F41 (2/694). In total, 55 of 453 (12.1%) isolates originating from suckling piglets with diarrhea could be classified as Enterotoxigenic *E. coli* (ETEC), with the combination of *faeG* (F4) and *elt* being the most common ETEC (38/55). *elt* could be detected in 70.2% of all *faeG*-positive ETEC isolates by PCR (Table 2).

Based on the combination of certain virulence genes, 43 out of 241 (17.8%) isolates from weaned piglets could be classified as ETEC. In this group, the combination of *faeG* (F4) and *elt* was also predominant (35/43), while *fasA* (F6), *fim41A* (F41), and *stb* were not detected in a single isolate. *AidA* was present in the majority of all *fedA* (F18) and *stx2*-positive isolates (12/14) (Table 3).

In general, *fedA* (F18) was numerically the most frequent virulence gene in all *aidA* positive ETEC isolates (13/14). *escV* was present in 10 isolates originating from suckling piglets and in 18 isolates from weaned piglets, respectively.

Most virulence genes associated with pathogenicity (*aidA*, *faeG, fasA, fedA, fim41A*, *elt, sta, astA, stx2*) were identified more frequently in isolates with hemolytic activity than in non-hemolytic *E. coli* (Table 4 and Table 5). The detection rate of *aidA* (p = 0.0353)*, faeG* (*p* <0.0001)*, fedA* (*p* = 0.0141)*, fim41A* (*p* = 0.0181)*, astA* (*p* = 0.0003)*, elt* (*p* <0.0001), and *sta* (*p* <0.0001) was significantly higher in isolates from suckling piglets displaying hemolytic activity than in isolates without hemolytic activity.

Virulence genes like *aidA* (*p* = 0.0362)*, faeG,* (*p* < 0.0001)*, fedA* (*p* < 0.0001)*, astA* (*p* = 0.0259)*, elt* (*p* < 0.0001)*, sta* (*p* = 0.0362), *and stx2* (*p* = 0.001) were detected significantly more often in isolates from weaned piglets displaying hemolytic activity than in those without hemolytic activity.

In addition, *cnf1* was exclusively found in isolates that were positive for *papC* (17/17), but only 27.0% of all *papC*-positive isolates were also positive for *cnf1*. The detection rate of *faeG* (F4) was significantly increased in *fimH* negative isolates (Standardized residual >1.96; *p* <0.05) but not vice versa (Standardized residual <1.96). The detection rate of *fedA* (F18) was decreased if the isolate was *fimH* negative (Standardized residual <1.96). Comparing the detection rate of *faeG* in different sampling materials, *faeG* was more frequently detected in isolates from fecal samples than in isolates from the intestines (*p* = 0.0116). The odds ratio of detecting *faeG* in *E. coli* isolates from fecal samples compared to isolates from intestinal samples was 2.732 (95% CI 1.304–5.721).

## 3. Discussion

Antimicrobial resistances were determined by agar disk diffusion test and not by determining distinct resistance encoding genes or resistance-mediating mutations. Furthermore, due to limitations of available clinical breakpoints towards most tested antimicrobials for porcine *E. coli*, clinical breakpoints for humans (Table 6) were applied for the interpretation of all twelve antimicrobial substances on disk diffusion tests instead. Therefore, antimicrobial resistance profiles should be interpreted with caution.

In the present study, resistance levels to ampicillin showed approximately the same level as in other European countries [19]. High resistance levels could be linked to the frequent use of β-lactam antibiotics to treat suckling-piglets against diseases like clostridiosis. This could also explain why resistance rates of isolates from suckling piglets to ampicillin were significantly higher than those from weaned piglets.

Tetracyclines are also more frequently applied to treat livestock in Austria than other antimicrobial substances [20]. Therefore, observed high resistance levels in our investigations might be the result of a widespread use of tetracyclines, as already assumed before [21]. In recent Spanish investigations using agar disk diffusion testing, resistances of clinical porcine *E. coli* isolates were also more frequently detected against amoxicillin and tetracycline than against ceftiofur, gentamicin, and enrofloxacin [22]. It can be assumed that resistances of porcine *E. coli* towards tetracyclines emerged, when antimicrobials were also used as growth promoters. Tetracyclines are not licensed for the treatment of diarrhea or edema disease in Austria but are mainly applied to treat respiratory diseases in fattening pigs.

Our standard microbiological diagnostic procedures do not test resistances to trimethoprim and sulfamethoxazole separately since both are only licensed in combination for the application in pigs. Resistance rates towards trimethoprim–sulfamethoxazole of porcine *E. coli* were relatively high in our investigations, which goes along with the results of other European surveys [19,23]. Similar findings were also reported in Germany, although the number of isolates with resistances to trimethoprim–sulfamethoxazole has slightly decreased over the past 14 years [24].

Chloramphenicol is prohibited as a therapy in livestock due to its toxic attributes [25]. Florfenicol though is commonly used to treat respiratory diseases in swine. In a Danish study, only 12% of all chloramphenicol resistant isolates were also resistant to florfenicol [26]. But even if the use of florfenicol in swine stocks is the reason of resistances to chloramphenicol in about 18.5% of all isolates, we have to keep in mind that in Austria florfenicol is not licensed to treat *E. coli* associated diseases. In our investigations, hemolytic isolates which were more frequently classified as ETEC had higher resistance rates to chloramphenicol than non-hemolytic isolates (Table 1).

An increase of gentamicin resistant porcine *E. coli* isolates may be attributed to the use of apramycin and other aminoglycosides in swine stocks since cross-resistances are reported [27]. Observed higher resistance rates of isolates with hemolytic activity to gentamicin and trimethoprim-sulfamethoxazole in comparison to non-hemolytic *E. coli* might be caused by the frequent use of both antibiotics to treat colibacillosis in the field. Noteworthy, virulence genes associated with pathogenicity were more frequently detected in isolates with hemolytic activity.

In our investigations, the probability of an isolate being resistant to ampicillin and trimethoprim-sulfamethoxazole was increased, if the isolate was resistant to ciprofloxacin. In trials, the number of ampicillin and trimethoprim-sulfamethoxazole resistant *E. coli* was increased after an intramuscular administration of ciprofloxacin [28]. In total, 16.4% of all isolates were resistant to ciprofloxacin, which is relatively high compared to investigations in other European countries [19]. In general, it can be assumed that the observed higher resistance levels to fluoroquinolones compared to cephalosporins in our investigations might be linked to their more frequent use in the treatment of *E. coli*-associated diseases in swine.

Observed resistances of *E. coli* to cephalosporins may reflect the wide spread of ESBL-producing *E. coli*, but could be also a potential result of their general use in Austrian swine stocks [20]. Out of all ceftazidime resistant isolates, 88.2% could be classified as ESBL-producing *E. coli*. However, only 30.9% of all isolates which were classified as ESBL-producing *E. coli* showed an in vitro resistance to ceftazidime. This could be due to the use of human clinical breakpoints, as well as using phenotypic methods to determine ESBL instead of testing genes.

Similar to ceftazidime, resistances to aztreonam were mainly observed in ESBL-producing isolates. However, compared to cephalosporins, aztreonam is not licensed to treat pigs in Austria. Despite ESBLs being commonly defined to confer bacterial resistance to penicillins and cephalosporins of the first, second, and third generation, as well as aztreonam, only 7.7% of our isolates that were classified as ESBL-producing were also resistant to aztreonam.

Although neither fosfomycin nor any related substance is licensed to treat pigs in the European Union, fosfomycin resistant isolates were observed in our investigations. Thus, swine might act as a reservoir for fosfomycin resistant *E. coli* in Austria. Isolates with plasmids carrying *fosA3* that encodes resistance to fosfomycin, as well as ESBL genes, have already been detected in livestock [29]. Nevertheless, in our study no ESBL-producing isolate was resistant to fosfomycin.

Neither meropenem- nor amikacin resistant *E. coli* were detected, both not licensed for use in animals in the EU. The absence of isolates being resistant to amikacin in swine stocks was also observed before [30], while Carbapenem resistant *E. coli* from European swine stocks have occasionally been detected [31], concretely in Germany [32,33] and Italy [34].

Data on the amount of used antibiotics by the different herd veterinarians were not available for this analysis. Those data could have helped to explain potential associations between the enhanced and blanket usage of certain antibiotics by each veterinarian. Interestingly, isolates from the herd veterinarian with the lowest resistance rates to tetracycline had the second highest resistance rates against ciprofloxacin, which could reveal the preference of certain veterinarians to use certain antibiotics.

The impact of certain virulence genes, like *astA, aidA*, and *cnf1*, might be higher than expected, whereas genes previously exclusively associated with pathogenicity like *sta* and *stb* were detected in only a few isolates from clinically diseased piglets. Furthermore, many of these virulence factors are not contained in commercial vaccines.

The absence of any investigated virulence gene in only a small number of all isolates (39/694) has also been observed in several other studies [35,36]. However, in those studies a PCR for *fimH*, which was the most frequently detected virulence gene in our investigation, was not applied. In an Australian investigation *fimH* was detected in all *E. coli* isolates from piglets with neonatal diarrhea [16], although an association between neonatal diarrhea and *fimH* has not been made so far. Nevertheless, the intestines are described to be the main reservoir of UPEC [37], which could explain the detection of *fimH* in so many isolates derived from feces or intestines.

The observed significantly increased detection rate of *faeG* in isolates negative for *fimH* could define *fimH* as an indicator of non-pathogenic *E. coli* colonizing the intestines. A correlation between the occurrence of *fimH* and *faeG* in porcine *E. coli* has not been described before. While *fimH* is a gene often associated with UPEC, F4 (*faeG*) are associated with ETEC. Thus, detection of both genes in the same isolate emphasizes the insufficiency of the classification of *E. coli* due to distinct virulence gene patterns. The occurrence of *fimH* and *faeG* in the same isolate has been observed in our investigations in several isolates.

*E. coli* isolates being positive for *faeG* were more likely to be detected in isolates from fecal samples compared to isolates from intestines. Therefore, fecal samples could be more adequate to detect *faeG* than intestinal samples.

Compared to previous studies, the number of isolates which were classified as ETEC out of all isolates from weaned piglets (43/241) was relatively low [35,36]. In accordance with results of other studies we detected *astA* more often in isolates which were positive for *elt* [9,38] and *faeG* (F4) was detected more frequently than *fanC* (F5), *fasA* (F6), and *fim41A* (F41) [35,36,38]. The low detection rates for *fanC*, *fasA*, and *fim41A* were explained to be the result of collecting samples only from piglets which were older than 14 days. However, recently no European study on the occurrence of virulence genes in *E. coli* isolates from younger suckling piglets has been published. However, in our study a high number of samples was also taken from suckling piglets with less than 14 days of age. Thus, we assume that a shift away from ETEC containing F5, F6, and F41 might have happened over the last decades.

Although the combination of *faeG* and *elt* was also noted frequently in our investigations, only one isolate was also positive for *stb*, whereas the combination of *faeG, elt*, and *stb* was observed most frequently in ETEC in other mentioned studies [36,38].

The impact of AIDA-I on the pathogenesis of PWD and ED is discussed controversially. However, it has already been proven that strains containing only *aidA* and *stb* were able to colonize the intestine, form biofilms, and induce diarrhea [12]. A strong linkage between *fedA* (F18) and *aidA* has already been described [39]. That result correlates well with our findings, in which the combination of *faeG* and *aidA* only occurred three times (twice together with *fedA*), whereas *fedA* and *aidA* were detected together in 13 isolates. In fact, *aidA* has already been discussed to play a significant role in the pathogenesis of ED [39].

Despite the fact that hemolytic activity is not a necessary attribute for pathogenic *E. coli*, it has been demonstrated several times that the majority of ETEC isolates from swine exhibits hemolytic activity on blood agar plates [38,40]. This correlates well with our investigations, in which several virulence genes associated with pathogenicity (*aidA, faeG, fasA, fedA, fim41A*, *elt, sta, astA, stx2*) were detected more frequently in hemolytic *E. coli* than in *E. coli* without hemolytic activity (Table 4,5). In addition, the occurrence of these genes was much lower in ESBL-producing isolates (Table 4,5), from which none showed hemolytic activity.

Data about the impact of *cnf1* in the pathogenesis of PWD is rare, while it has been described to be a typical gene of Necrotoxic *E. coli* (NTEC), along with genes encoding P and S fimbriae (*papC, sfa*) [18]. The correlation between the occurrence of *cnf1* and *papC* has also been observed before [18]. Therefore, the role of several described genes in the pathogenesis of colibacillosis should not be underestimated.

## 4. Materials and Methods

### 4.1. Sample Collection

Between 2016 and 2018, 694 *E. coli* strains were isolated from feces and intestines derived from clinically diseased suckling and early weaned piglets from Austrian swine stocks in the course of routine diagnostics performed at the Institute for Microbiology, University of Veterinary Medicine Vienna (Appendix A). *E. coli* isolates were identified using standard microbiological procedures. Animals were selected by the herd veterinarian based on the presence of typical clinical symptoms of neonatal diarrhea, postweaning diarrhea, or edema disease, since the aim of the vet was to obtain results of antimicrobial susceptibility testing of the causative pathogens for the choice of an appropriate treatment. Intestines (n = 312) and feces (n = 372) were collected from 215 different farms. Depending on the veterinarian, the number of farms by herd veterinarian from which specimens were taken varied between one and 21.

### 4.2. Antimicrobial Susceptibility Testing

Susceptibility testing was performed by agar disk diffusion according to the recommendations given by the CLSI document M100 (28th ed.) (Clinical and Laboratory Standards Institute (CLSI), 2018). Human clinical breakpoints (M100-28) were applied for the interpretation of zone diameters of all tested antimicrobial substances (Table 6). Isolates were classified as ESBL-producing if they showed an increase of over 5 mm in a zone diameter for either ceftazidime or cefotaxime tested in combination with clavulanate compared to the zone diameter of ceftazidime or cefotaxime alone in agar disk diffusion (Becton Dickinson, Heidelberg, Germany).

### 4.3. Virulence Genes

The isolates were examined by PCR targeting 22 genes (*fimH, aidA, faeG, fanC, fasA, fedA, fim41A, elt, sta, stb, astA, stx1, stx2, invE, aggR, pic, ent, papC, iucD, cnf1, escV, bfpB*) as previously described [16,41,42,43].

### 4.4. Evaluation

All results were summarized retrospectively using TIS® (Tierspitalinformationssystem Orbis VetWare, Agfa HealthCare, Bonn, Germany) and Microsoft Excel. In total, 453 isolates were recovered from suckling piglets and 241 from weaned piglets. Overall, 403 isolates were characterized as non-hemolytic *E. coli*, 142 as hemolytic *E. coli* and 166 isolates as ESBL-producing *E coli* without hemolytic activity. Due to the relevance of antimicrobials in public health and/or their importance for therapeutic use in swine and humans, resistance patterns to the following twelve antimicrobial substances or combinations were analyzed: ampicillin, tetracycline, trimethoprim/sulfamethoxazole, chloramphenicol, gentamicin, tobramycin, ciprofloxacin, ceftazidime, aztreonam, fosfomycin, amikacin, and meropenem (Becton Dickinson, Heidelberg, Germany).

### 4.5. Statistical Analyses

The prevalence of virulence genes and antimicrobial resistance in dependence of the age (suckling piglets and weaned piglets), as well as the phenotype of *E. coli* (non-hemolytic *E. coli,* hemolytic *E. coli*, ESBL *E. coli*), was assessed via Microsoft Excel. Furthermore, Pearson Chi Squared and Fisher’ Exact Tests were calculated using the R statistical computing environment (R Core Team R: A language and environment for statistical computing. R foundation for statistical computing, Vienna, Austria, version 3.5.3) to verify potential dependencies between 1.: ampicillin, ciprofloxacin, and trimethoprim-sulfamethoxazole; 2.: *fimH* and *faeG*; 3.: *papC* and *cnf1*, as well as 4.: the detection rate of *faeG* depending on the sampling material; 5.: the detection rates of virulence genes *(aidA, faeG, fedA, fim41A, elt, sta, astA, stx2)* depending on the presence of hemolytic activity; and 6.: considering the age group and clinical symptoms, the influence of the herd veterinarian on antimicrobial resistance rates was investigated using different generalized linear models.

We aimed to reduce the models to include only the most relevant factors and applied the Akaike information criterion (AIC) to determine the best model. The AIC uses the parsimony principle to reduce the number of factors considered in the model and incorporates both the complexity of the estimated model and their associated goodness-of-fit to the data [44]. In detail, first model analyzed the influence of age group, clinical symptoms and herd veterinarian on a binary outcome (defined as no detected resistance within the tested antibiotics and at least one detected resistance), after application of AIC only the age group was kept in the model. The second model is similar, with a slightly different binary outcome (defined as less than three detected resistances within the tested antibiotics and at least three detected resistances), after application of AIC only age group and herd veterinarian were kept in the model. The third model used the relative amount of resistances within the tested antibiotics as metric outcome; all three variables were kept in the model after using stepwise AIC. A P-values of <0.05 were considered as significant. The used R packages were “gmodels”, “oddsratio”, “MASS” and “ggplot2” [45,46,47].

## 5. Conclusions

Resistance levels to ciprofloxacin are higher than previously reported. Resistances to substances that are not applied to swine stocks like fosfomycin emphasize the potential of transmission between different species. Hemolytic activity and absence of *fimH* are potential indicators for the pathogenicity of *E. coli* but not necessary attributes. Furthermore *aidA, astA*, and *cnf1* might play a key role in the pathogenesis of colibacillosis, but further investigations are needed.

## Figures and Tables

**Figure 1 antibiotics-09-00208-f001:**
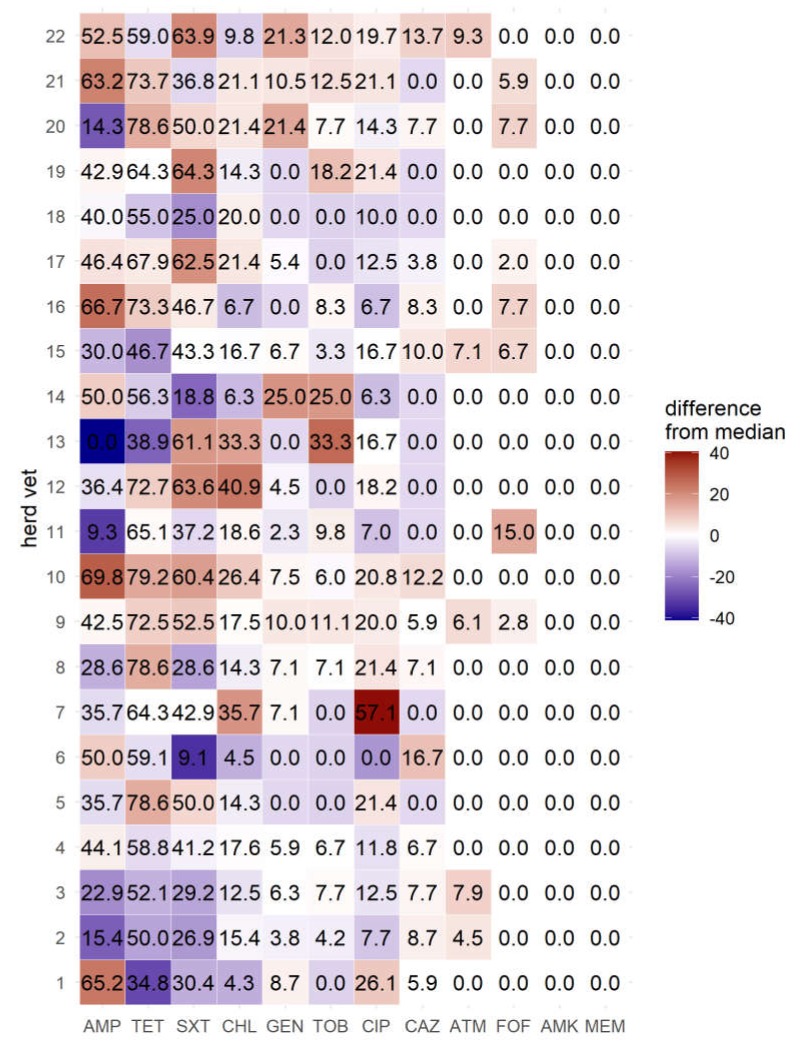
Frequency of resistant *E. coli* isolates by herd veterinarian (in %). AMP = ampicillin; TET = tetracycline; SXT = trimethoprim-sulfamethoxazole; CHL = chloramphenicol; CIP = ciprofloxacin; GEN = gentamicin; TOB = tobramycin; CAZ = ceftazidime; ATM = aztreonam; FOF = fosfomycin; AMK = amikacin; MEM = Meropenem.

**Table 1 antibiotics-09-00208-t001:** Frequency of resistant *E. coli* isolates.

	Non-Hemolytic *E. coli*	Hemolytic *E. coli*	Non-Hemolytic ESBL-Producing *E. coli*	All *E. coli* Isolates
**Ampicillin**	66.3%	58.2%	100.0%	71.9%
**Tetracycline**	67.5%	63.4%	71.9%	67.7%
**Trimethoprim-Sulfamethoxazole**	3.2%	45.4%	63.0%	49.5%
**Chloramphenicol**	16.9%	20.8%	19.9%	18.5%
**Gentamicin**	4.0%	8.4%	15.4%	7.7%
**Tobramycin**	4.2%	6.6%	10.9%	6.2%
**Ciprofloxacin**	14.3%	10.0%	27.2%	16.4%
**Ceftazidime**	1.2%	0.0%	30.9%	5.9%
**Aztreonam**	0.3%	0.0%	7.7%	2.2%
**Fosfomycin**	2.8%	2.5%	0.0%	2.0%
**Amikacin**	0.0%	0.0%	0.0%	0.0%
**Meropenem**	0.0%	0.0%	0.0%	0.0%

Percentage of resistant isolates out of all isolates (last column) and out of each phenotypic group towards twelve antimicrobial substances. ESBL = extended-spectrum β-lactamases.

**Table 2 antibiotics-09-00208-t002:** Frequency of adhesion and toxin gene combinations (suckling piglets).

	***Elt***	***Sta***	***AstA***	***Elt, Sta***	***Elt, Stb***	***Elt, AstA***	***Sta, AstA***	***Elt, Sta, AstA***
***FaeG* (F4)**	17	3	5	3	1	14	2	1
***FanC* (F5)**	2		1					
***FasA* (F6)**		1						
***FaeG* (F4), *AidA***				1		1		
***FasA* (F6), *AidA***					1			
***FedA* (F18)*, AidA***		2						

Number of *E. coli* isolates originating from suckling piglets with diarrhea that were positive for an adhesion gene, as well as a toxin gene.

**Table 3 antibiotics-09-00208-t003:** Frequency of adhesion and toxin gene combinations (weaned piglets).

	*Elt*	*Sta*	*AstA*	*Elt, Sta*	*Elt, AstA*	*Sta, AstA*	*Stx2*
***FaeG* (F4)**	13	1		6	14		
***FanC* (F5)**			1				
***FedA* (F18)**	1	2			2	1	2
***FaeG* (F4), *AidA***					1		
***FedA* (F18)*, AidA***							11
***FaeG* (F4)*, FedA* (F18)*, AidA***					1		1

Number of *E. coli* isolates originating from weaned piglets with diarrhea or suspected edema disease that were positive for an adhesion gene, as well as a toxin gene.

**Table 4 antibiotics-09-00208-t004:** Frequency of virulence genes in different phenotypes (suckling piglets).

	n	*AidA*	*FaeG*	*FanC*	*FasA*	*FedA*	*Fim41A*	*Elt*	*Sta*	*Stb*	*AstA*
**Non-Hemolytic** ***E. coli***	264	3.0%	8.0%	2.3%	0.8%	1.5%	0.0%	3.8%	1.5%	3.4%	10.2%
**Hemolytic** ***E. coli***	58	8.6%	69.0%	0.0%	1.7%	6.9%	3.4%	51.7%	17.2%	1.7%	31.0%
**Non Hemolytic** **ESBL-Producing** ***E. coli***	122	2.5%	2.5%	0.0%	0.0%	0.0%	0.0%	4.1%	0.8%	1.6%	15.6%

Percentage of *E. coli* containing each virulence gene of all isolates from suckling piglets with diarrhea.

**Table 5 antibiotics-09-00208-t005:** Frequency of virulence genes in different phenotypes (weaned piglets).

	*n*	*AidA*	*FaeG*	*FedA*	*Elt*	*Sta*	*AstA*	*Stx2*
**Non-Hemolytic** ***E. coli***	121	16.5%	15.7%	7.4%	9.9%	3.3%	14.0%	4.1%
**Hemolytic *E. coli***	77	27.3%	33.8%	36.4%	35.1%	9.1%	22.1%	15.6%
**Non-Hemolytic ESBL–Producing *E. coli***	39	5.1%	7.7%	2.6%	12.8%	5.1%	10.3%	0.0%

Percentage of *E. coli* containing each virulence gene of all isolates from piglets affected by PWD or ED.

**Table 6 antibiotics-09-00208-t006:** Concentrations and breakpoints used for the interpretation of disk diffusion test.

Antibiotic	Concentration (µg/mg)	Breakpoint (mm)	
		S	R
**Ampicillin**	10	≥17	≤13
**Tetracycline**	30	≥15	≤11
**Trimethoprim-Sulfamethoxazole**	1.2523.75	≥16	≤10
**Chloramphenicol**	30	≥18	≤12
**Gentamicin**	10	≥15	≤12
**Tobramycin**	10	≥15	≤12
**Ciprofloxacin**	5	≥21	≤15
**Ceftazidime**	30	≥21	≤17
**Aztreonam**	30	≥21	≤17
**Fosfomycin**	200	≥16	≤12
**Amikacin**	30	≥17	≤14
**Meropenem**	10	≥23	≤19

Reference: Clinical and Laboratory Standards Institute (CLSI) M100 (28th ed.).

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
