# Peer review of "Prevalence of Virulence Genes and Antimicrobial Resistances in E. coli Associated with Neonatal Diarrhea, Postweaning Diarrhea, and Edema Disease in Pigs from Austria"

_antibiotics, 2020, doi:10.3390/antibiotics9040208_

Round 1
Reviewer 1 Report
Lines 22-23: Overall, 71.9%, 67.7% and 49.5% of all isolates were resistant to ampicillin, tetracycline and trimethoprim-sulfamethoxazole, respectively.
Reviewer: No mention is made about gentamicin resistance here, although this is shown in the results.
Lines 77-78: Consequently, the present study mainly aimed to investigate the frequency of certain 77 virulence genes in E. coli isolates from piglets displaying diarrhoea or clinical signs of ED in Austria.
Reviewer: One of the major weak points in this study is that healthy piglets have not been used as a control. Clinical manifestation of infection may not be present in all animals investigated owing to genetic variation, particularly since many genes are being analysed. In order to reduce this variability, the choice of animal must be based on specific genetic markers associated with ETEC susceptibility. An easier way to test that the reported patterns are indeed correlated with disease would be to test asymptomatic and/or healthy animals in the herd.
Did the herds investigated have a history of outbreaks? What were the management practices at the farm? Animal selection is not mentioned in the methods lines 320-23. This is a serious drawback; a list of criteria in the context of fimbrial genes is presented in Luise et al. Journal of Animal Science and Biotechnology (2019) 10:53. https://doi.org/10.1186/s40104-019-0352-7. Some of the criteria surely apply to the current study.
Line 139: In general, fedA (F18) was numerously the most frequent virulence gene in all aidA positive ETEC.
Reviewer: Should be “numerically”.
Lines 236-237: Thus, swine might act as a reservoir 236 for fosfomycin resistant E. coli in Austria.
Reviewer: This is an important finding, which should be mentioned in the abstract
Author Response
Point 1:
Lines 22-23: Overall, 71.9%, 67.7% and 49.5% of all isolates were resistant to ampicillin, tetracycline and trimethoprim-sulfamethoxazole, respectively.
Reviewer: No mention is made about gentamicin resistance here, although this is shown in the results.
Response 1:
Since gentamicin is applied frequently in swine stocks to treat colibacillosis and resistances were only observed in less than 10 % of all isolates, it was an excellent idea to include our results about gentamicin in the abstract as well. Information has been added in lines 23 – 24:
..while resistance levels to gentamicin and fosfomycin were 7.7 % and 2.0 %, respectively
Point 2:
Lines 77-78: Consequently, the present study mainly aimed to investigate the frequency of certain 77 virulence genes in E. coli isolates from piglets displaying diarrhoea or clinical signs of ED in Austria.
Reviewer: One of the major weak points in this study is that healthy piglets have not been used as a control. Clinical manifestation of infection may not be present in all animals investigated owing to genetic variation, particularly since many genes are being analysed. In order to reduce this variability, the choice of animal must be based on specific genetic markers associated with ETEC susceptibility. An easier way to test that the reported patterns are indeed correlated with disease would be to test asymptomatic and/or healthy animals in the herd.
Response 2:
You are absolutely right that not having included a control group is a big weak point. Unfortunately, the study design has been developed after samples were taken, since we decided to use data we have collected over the past three years about porcine E. coli that were isolated in the course of routine diagnostics. Furthermore, in the course of routine diagnostics we only receive specimens from clinically diseased animals, since herd veterinarians depend on antimicrobial susceptibility testing of isolates deriving from diseased pigs. Because big data was provided from lots of isolates deriving from many farms all over the country, it would have been a big waste not to use these data about virulence genes and antimicrobial resistances.
However, the aim of the study was to display the approximate prevalence of virulence genes associated with ETEC and antimicrobial resistances in clinically affected swine stocks, rather than performing a case-control study.
Most other cited studies reporting from the occurrence of virulence genes in porcine E. coli isolates also did not implement a control group as well, such as:
Brand, P.; Gobeli, S.; Perreten, V. Pathotyping and antibiotic resistance of porcine enterovirulent Escherichia coli strains from Switzerland (2014–2015). SAT 2017, 159, 373–380, doi:10.17236/sat00120.
Luppi, A.; Gibellini, M.; Gin, T.; Vangroenweghe, F.; Vandenbroucke, V.; Bauerfeind, R.; Bonilauri, P.; Labarque, G.; Hidalgo, Á. Prevalence of virulence factors in enterotoxigenic Escherichia coli isolated from pigs with post-weaning diarrhoea in Europe. Porc Health Manag 2016, 2, 20, doi:10.1186/s40813-016-0039-9.
Frydendahl, K. Prevalence of serogroups and virulence genes in Escherichia coli associated with postweaning diarrhoea and edema disease in pigs and a comparison of diagnostic approaches. Veterinary Microbiology 2002, 85, 169–182, doi:10.1016/S0378-1135(01)00504-1.
Point 3:
Did the herds investigated have a history of outbreaks? What were the management practices at the farm?
Response 3:
Although we have a lot of data, data if farms have a history of outbreaks was unfortunately not provided. To include management practices as well is certainly a very good idea, since management practices are known to play a very important role in the pathogenesis of either disease as well. Right at the moment, we are having a project in which a questionnaire is applied to get access to many data about management practices and feeding management in regard to PWD. For this study it would have been too much data, since we really wanted to focus on the prevalence of virulence genes and antimicrobial resistances in isolates from animals with diarrhea.
Point 4:
Animal selection is not mentioned in the methods lines 320-23. This is a serious drawback; a list of criteria in the context of fimbrial genes is presented in Luise et al. Journal of Animal Science and Biotechnology (2019) 10:53. https://doi.org/10.1186/s40104-019-0352-7. Some of the criteria surely apply to the current study.
Response 4:
Thank you very much, for reminding us to include the methods of animal selection. They were added to the lines 309-315:
Animals were selected by the herd veterinarian based on the presence of typical clinical symptoms of neonatal diarrhoea, postweaning diarrhoea or oedema disease, since the aim of the vet was to obtain results of antimicrobial susceptibility testing of the causative pathogens for the choice of appropriate treatment. Intestines (n = 312) and faeces (n = 372) were collected from 215 different farms. Depending on the herd veterinarian, the number of farms from which specimens were taken varied between one and 21.
Point 5:
Line 139: In general, fedA (F18) was numerously the most frequent virulence gene in all aidA positive ETEC.
Reviewer: Should be “numerically”.
Response 5:
Thank you. The sentence was changed as suggested in line 150:
In general, fedA (F18) was numerically the most frequent virulence gene in all aidA positive ETEC isolates (13/14).
Point 6:
Lines 236-237: Thus, swine might act as a reservoir 236 for fosfomycin resistant E. coli in Austria.
Reviewer: This is an important finding, which should be mentioned in the abstract
Response 6:
Thank you very much for the suggestion of mentioning that we could detect E. coli isolates that were resistant to fosfomycin, especially since neither fosfomycin nor related antimicrobials are applied in Austrian swine stocks. Due to limitations of 200 words in the abstract, we would only consider to present results but not further thoughts about fosfomycin resistant E. coli in Austrian swine stocks in the abstract. Changes were in lines 23 & 24:
while resistance levels to gentamicin and fosfomycin were 7.7 % and 2.0 %, respectively .
Reviewer 2 Report
In this study, the authors screen phenotypic antimicrobial resistance and presence of virulence genes in Escherichia coli isolates collected from diseased pigs in Austria, and try to establish a relationship between the traits and the ability to promote disease in piglets.
Main comments:
The statement in lines 54-56 needs a reference.
In the material and methods, a statistical analyses section is included. However, statistic results are not presented anywhere in the whole text, with the exception of the odds-ratio in line 315. The authors do not show if their observations are statistically significant or not.
Data presented at table 2 (and possibly table 1 too) would be better perceived in graphics rather than in a table.
The discussion is extremely long. Several information is added without a clear link to the results of the study. Often the authors associate the observed resistance to possible genes. However, they did not screen any resistance genes, making these observations just assumptions.
Two different approaches were taken for detection of resistance and virulence genes. The former is a phenotypic detection and the latter a genotypic one. Resistance to antimicrobials can be associated with several resistance genes, while presence of the virulence gene does not mean expression of that gene.
The tables mentioned in the section “Supplementary Materials” are not supplementary, are the tables inserted in the main text. On the other site, the table submitted as supplementary is not mentioned here.
Author Response
n this study, the authors screen phenotypic antimicrobial resistance and presence of virulence genes in Escherichia coli isolates collected from diseased pigs in Austria, and try to establish a relationship between the traits and the ability to promote disease in piglets.
Main comments:
Point 1:
The statement in lines 54-56 needs a reference.
Response 1:
We thank the reviewer for the suggestion to add a reference.
We inserted the reference at the end of the sentence of lines 54-58.:
Epithelial adherence is predominantly facilitated by the adhesive fimbriae F4 (faeG) and F18 (fedA), as well as by F5 (fanC), F6 (fasA) and F41 (fim41A). The detection of at least one enterotoxin gene (elt, sta, stb) together with one gene coding for fimbriae including F4, F5, F6, F18 and F41 in a single E. coli isolate is defined as an essential criterion for the classification of porcine ETEC [8].
The following reference was added:
Fairbrother, J.M.; Nadeau, É.; Gyles, C.L. Escherichia coli in postweaning diarrhea in pigs: an update on bacterial types, pathogenesis, and prevention strategies. Anim. Health. Res. Rev. 2005, 6, 17–39, doi:10.1079/AHR2005105.
Point 2:
In the material and methods, a statistical analyses section is included. However, statistic results are not presented anywhere in the whole text, with the exception of the odds-ratio in line 315. The authors do not show if their observations are statistically significant or not.
Response 2:
We thank the reviewer on reminding us to present statistic results as well including p-values. We totally agree, that statistic results should be presented with calculated P-Values. P-Values, Standard-Residuals or Calculated odds-ratios were added to the results, as mentioned in materials & methods, for dependencies between
- ciprofloxacin and ampicillin or trimethoprim-sulfamethoxazole respectively in lines 93-95:
Isolates that were resistant to ciprofloxacin were more likely to be resistant to ampicillin (OR = 5.381 [95% CI 2.855 - 10.140]) and trimethoprim-sulfamethoxazole (OR = 3.482 [95% CI 2.554 – 4.747]).
- the age group and resistance rates in lines 108-109:
Weaned piglets had significant lower resistance rates to tested antibiotics in all three calculated generalized linear models (Model 1: p = 0.0107, Model 2: p = 0.0397, Model 3: p = 0.0019).
- detected virulence genes and haemolytic activity in isolates from suckling piglets in lines 155-157:
The detection rate of aidA (p = 0.353), faeG (p <0.0001), fedA (p = 0.0141), fim41A (p = 0.0181), astA (p = 0.0003), elt (p <0.0001) and sta (p <0.0001) was significantly higher in isolates from suckling piglets displaying haemolytic activity than in isolates without haemolytic activity.
- detected virulence genes and haemolytic activity in isolates from weaned piglets in lines 162-164:
Virulence genes like aidA (p = 0.0362), faeG, (p <0.0001), fedA (p <0.0001), astA (p = 0.0259), elt (p <0.0001), sta (p = 0.0362) and stx2 (p = 0.001) were detected significantly more often in isolates from weaned piglets displaying haemolytic activity than in those without haemolytic activity.
- fimH and faeG
- the detection rate of faeG depending on the sample material, both in lines 171-176.:
The detection rate of faeG (F4) was significantly increased in fimH negative isolates (Standardised residual >1.96; p <0.05) but not vice versa (Standardised residual <1.96). The detection rate of fedA (F18) was decreased if the isolate was fimH negative (Standardised residual <1.96). Comparing the detection rate of faeG in different sampling material, faeG was more frequently detected in isolates from faecal samples than in isolates from the intestines (p = 0.0116). The odds ratio of detecting faeG in E. coli isolates from faecal samples compared to isolates from intestinal samples was 2.7 (95% CI).
Point 3:
Data presented at table 2 (and possibly table 1 too) would be better perceived in graphics rather than in a table.
Response 3:
The authors thank the reviewer for the suggestion to better use a figure instead of a table for the visualization of antimicrobial resistances. We are totally aware that, we have already used probably too many tables. Table 2 has been replaced by a Figure 1 between Lines 117-122.
Figure 1. Frequency of resistant E. coli isolates by herd veterinarian (in %)
Percentage of resistant E. coli isolates by the herd veterinarian; AMP = ampicillin; TET = tetracycline; SXT = trimethoprim-sulfamethoxazole; CHL = chloramphenicol; CIP = ciprofloxacin; GEN = gentamicin; TOB = tobramycin; CAZ = ceftazidime; ATM = aztreonam; FOF = fosfomycin; AMK = amikacin; MEM = Meropenem
Point 4a:
The discussion is extremely long.
Response 4a:
Thanks for this comment, the discussion has been shortened from 2211 to 1729 words.
Point 4b:
several information is added without a clear link to the results of the study.
Response 4b:
The authors thank the reviewer for this comment.
References to our results of antimicrobial resistances are now mentioned in the discussion in lines 207-209:
But even if the use of florfenicol in swine stocks is the reason of resistances to chloramphenicol in about 18.5% of all isolates, we have to keep in mind that in Austria florfenicol is not licensed to treat E. coli associated diseases.
And 221-222:
In total, 16.4% of all isolates were resistant to ciprofloxacin, which is relatively high compared to investigations in other European countries [19].
Point 4c:
Often the authors associate the observed resistance to possible genes. However, they did not screen any resistance genes, making these observations just assumptions.
Response 4c:
We deleted of references to studies in which resistance genes were discussed and replaced them with studies and surveys in which disk diffusion test was applied.
The following references have been added instead in lines
191-194:
In recent Spanish investigations using agar disk diffusion, resistances of clinical porcine E. coli isolates were also more frequently detected against amoxicillin and tetracycline than against ceftiofur, gentamicin and enrofloxacin [22].
And 202-204:
Similar findings were also reported in Germany, although the number of isolates with resistances to trimethoprim-sulfamethoxazole has slightly decreased over the past 14 years [24].
The following reference was added
- Aguirre, L.; Vidal, A.; Seminati, C.; Tello, M.; Redondo, N.; Darwich, L.; Martín, M. Antimicrobial resistance profile and prevalence of extended-spectrum beta-lactamases (ESBL), AmpC beta-lactamases and colistin resistance (mcr) genes in Escherichia coli from swine between 1999 and 2018. Porc Health Manag 2020, 6, 8, doi:10.1186/s40813-020-00146-2
Point 5:
Two different approaches were taken for detection of resistance and virulence genes. The former is a phenotypic detection and the latter a genotypic one. Resistance to antimicrobials can be associated with several resistance genes, while presence of the virulence gene does not mean expression of that gene.
Response 5:
You are absolutely right. We are aware that this is a limitation of the present study. On the other hand, we kindly ask to explain why we did not perform genotypic susceptibility testing.
During the routine diagnostic of porcine E. coli isolates a phenotypic susceptibility testing as well as screening for virulence factors are always performed. Unfortunately, genotypic susceptibility testing is not a part of routine diagnostic protocols in our lab. Since the present study is retrospective, we are not able to perform genotypic susceptibility testing because neither strains nor their DNA were stored for such a long period of time.
Since, we are aware that these are two completely different approaches, detection of virulence genes and resistances were not compared in this study and separated as two different parts throughout the course of the whole manuscript.
Point 6:
The tables mentioned in the section “Supplementary Materials” are not supplementary, are the tables inserted in the main text. On the other site, the table submitted as supplementary is not mentioned here.
Response 6:
Thank you for reminding us to fix this. The names of Supplementary Materials are updated in lines 376 and 377.
Reviewer 3 Report
This is an interesting manuscript, describing the prevalence of antimicrobial resistance ans the prevalence of virulnece factors. But there are some questions to the authors:
1.- Why did the authors decide to separate the strains in haemolytic and non haemolytic? are there any association between haemolysis and antimicrobial resistance? or between haemolysis ans virulence? The authors need to clarify this division, and discuss about that. Needs statistical analysis.
2- The authors write in line 103 "Weaned piglets had significant lower resistance rates to tested antibiotics in all three calculated statistical models". Which are these models, where are they described?
3- line 106-107 " However, considering age group and clinical symptoms, the data show no significant influence of the herd veterinarian on the resistance profiles in any of the models." where are the statistical? the authors only presented percentatges.
Regarding the Herd veterinarian, there is no description about that in the manuscript, are the herds in only one farm? are there some farms?. This needs a deep description.
4.-Finally, as farm animals generally are a reservoir of antimicrobial resistant bacteria, it would be nice to analyse the MLST of the ESBL-E.coli, to see if they are epidemiologically related with Human isolates.
Author Response
Point 1:
1.- Why did the authors decide to separate the strains in haemolytic and non haemolytic? are there any association between haemolysis and antimicrobial resistance? or between haemolysis ans virulence? The authors need to clarify this division, and discuss about that.
Response 1:
We thank the reviewer for this comment. In routine bacteriology, haemolysis on blood agar is one the most critical cultural characteristic of an isolate. Thus, separation of haemolytic and non haemolytic isolates is always proceeded in the course of routine diagnostics at the Institute of Microbiology at the University of Veterinary Medicine in Vienna. Therefore, data about the colony morphology (i.e. haemolysis vs. no haemolysis) of E. coli was already available. Since haemolytic activity is described as an indicator for pathogenicity of E. coli, (so the frequent detection of virulence genes) (references below: Fairbrother and Nadeau) we were keen on separating all isolates into those with haemolytic and those without haemolytic activity. We added details about this topic (lines 64-67):
Haemolytic activity in ETEC from weaned piglets is observed in most isolates containing faeG and all isolates containing fedA. Thus, haemolytic activity could act as an indicator for pathogenicity, although its impact should not be overestimated [14].
[14] Fairbrother, J.M.; Nadeau, E. Colibacillosis. In Diseases of swine, 11th ed.; Zimmerman, J.J., Karriker, L.A., Ramirez, A., Schwartz, K.J., Stevenson, G.W., Zhang, J., Eds.; Wiley-Blackwell: Hoboken, NJ, 2019; pp. 807–834.
Since data is mainly referring to E. coli from weaned piglets but not from suckling piglets, we were interested to see if virulence genes can also be detected more frequently in haemolytic E.coli from suckling piglets.
However, there is no data about the occurrence of antimicrobial resistances depending on haemolytic activity. Therefore, we wanted to further investigate if resistances to antimicrobial substances might depend as well on the haemolytic activity of E. coli.
Point 1a:
Needs statistical analysis.
Response 1a:
The authors thank the reviewer for the comment to also include the analysis if virulence genes were detected significantly more often in isolates with haemolytic activity. Information was added in lines 155-157:
The detection rate of aidA (p = 0.353), faeG (p <0.0001), fedA (p = 0.0141), fim41A (p = 0.0181), astA (p = 0.0003), elt (p <0.0001) and sta (p <0.0001) was significantly higher in isolates from suckling piglets displaying haemolytic activity than in isolates without haemolytic activity.
and 162-164:
Virulence genes like aidA (p = 0.0362), faeG, (p <0.0001), fedA (p <0.0001), astA (p = 0.0259), elt (p <0.0001), sta (p = 0.0362) and stx2 (p = 0.001) were detected significantly more often in isolates from weaned piglets displaying haemolytic activity than in those without haemolytic activity.
Point 2:
2- The authors write in line 103 "Weaned piglets had significant lower resistance rates to tested antibiotics in all three calculated statistical models". Which are these models, where are they described?
Response 2:
The authors thank the reviewer for reminding us to add information on the applied statistical methods in particular for the results of different age groups. Information was added in lines 355-366:
We aimed to reduce the models to include only the most relevant factors and applied the Akaike information criterion (AIC) to determine the best model. The AIC uses the parsimony principle to reduce the number of factors considered in the model and incorporates both the complexity of the estimated model and their associated goodness-of-fit to the data [44]. In detail, first model analysed the influence of age group, clinical symptoms and herd veterinarian on a binary outcome (defined as no detected resistance within the tested antibiotics and at least one detected resistance), after application of AIC only the age group was kept in the model. The second model is similar, with a slightly different binary outcome (defined as less than three detected resistances within the tested antibiotics and at least three detected resistances), after application of AIC only age group and herd veterinarian were kept in the model. The third model used the relative amount of resistances within the tested antibiotics as metric outcome; all three variables were kept in the model after using stepwise AIC. A P-values of <0.05 were considered as significant.
44. Pinior, B.; Firth, C.L.; Loitsch, A.; Stockreiter, S.; Hutter, S.; Richter, V.; Lebl, K.; Schwermer, H.; Käsbohrer, A. Cost distribution of bluetongue surveillance and vaccination programmes in Austria and Switzerland (2007–2016). Veterinary Record 2018, 182, 257–257, doi:10.1136/vr.104448
Point 3:
3- line 106-107 " However, considering age group and clinical symptoms, the data show no significant influence of the herd veterinarian on the resistance profiles in any of the models." where are the statistical? the authors only presented percentatges.
Response 3:
The authors thank the reviewer for this statement. P-Values for differences between the age group were added in lines 108 & 113:
Weaned piglets had significantly lower resistance rates to tested antibiotics in all three calculated generalized linear models (Model 1: p = 0.0107, Model 2: p = 0.0397, Model 3: p = 0.0019). Examined isolates originating from swine stocks supervised by herd veterinarian 1 exhibiting the lowest resistant rates against tetracycline, had the second highest resistance rates to ciprofloxacin (Figure 1). However, considering age group and clinical symptoms, the data show no significant influence of the herd veterinarian on the resistance profiles in any of the generalized linear models.
Since the herd veterinarian had no significant influence on resistance profiles in any of the models, P-Values were not added into the results.
Point 3a:
Regarding the Herd veterinarian, there is no description about that in the manuscript, are the herds in only one farm? are there some farms?. This needs a deep description.
Response 3a:
Thank you very much for reminding us to include how many farms were investigated by each herd veterinarian. All farms were individual, mostly family owned swine herds which is due to the small scale structure of the swine industry in Austria.
Because a complete list with the number of isolated E. coli by farm and number of farms by herd veterinarian would be too complex and confusing, we decided to not include it to the manuscript. The range of investigated farms per herd veterinarian was included in lines 313-315:
Intestines (n = 312) and faeces (n = 372) were collected from 215 different farms. Depending on the herd veterinarian the number of farms, from which specimens were taken varied between one and 21.
Point 4:
4.-Finally, as farm animals generally are a reservoir of antimicrobial resistant bacteria, it would be nice to analyse the MLST of the ESBL-E.coli, to see if they are epidemiologically related with Human isolates.
Response 4:
This is an excellent comment, since it would emphasize the impact of porcine ESBL – E. coli to public health even more. We are aware that our study had limitations due to its retrospective design. So, no ESBL – producing E. coli were preserved, because at the time point of study design isolates were no longer available. Nevertheless, we thank the reviewer for this idea and are considering to analyse the MLST of porcine ESBL – producing E. coli in the near future.
Round 2
Reviewer 1 Report
I think the authors have addressed most of my concerns.
Reviewer 2 Report
Response 1: The detection of at least one enterotoxin gene (elt, sta, stb) together with one gene coding for fimbriae including F4, F5, F6, F18 and F41 in a single E. coliisolate is defined as an essential criterion for the classification of porcine ETEC [8].
The following reference was added:
Fairbrother, J.M.; Nadeau, É.; Gyles, C.L. Escherichia coli in postweaning diarrhea in pigs: an update on bacterial types, pathogenesis, and prevention strategies. Anim. Health. Res. Rev. 2005, 6, 17–39, doi:10.1079/AHR2005105.
Comment: I don’t see in the added reference the affirmation that it is enough to detect one enterotoxin and one fimbriae gene to classify one E, coli isolate as porcine ETEC.
Response 2: 3. detected virulence genes and haemolytic activity in isolates from suckling piglets in lines 155-157:
The detection rate of aidA (p = 0.353), faeG (p <0.0001), fedA (p = 0.0141), fim41A (p = 0.0181), astA (p = 0.0003), elt (p <0.0001) and sta (p <0.0001) was significantly higher in isolates from suckling piglets displaying haemolytic activity than in isolates without haemolytic activity.
Comment: The result for aidA is not statistically significant.
Response 6: Thank you for reminding us to fix this. The names of Supplementary Materials are updated in lines 376 and 377.
Comment: I don’t see Figure S1 in submitted files (for me, it seems that, again, the authors are including here as supplementary a figure that is in the main document).
New comments:
Line 230 – “described” should be replaced by “Phenotypic”
Line 326 - Why did the authors include table 6 with resistance breakpoints? If they followed the CLSI guidelines, there is no point in including an extra table here (maximum they can include it as supplementary). By the way, why did the authors use the American guidelines instead of the European ones, once the data reported is from an European country?
Breakpoint should be written together as a whole word, not 2 separate words.
In line 237 it is used “EU” while in line 243 it is written “European Union”. The order should be the opposite.
Reviewer 3 Report
The authors addressed my suggestions.